# Influence of Natural and Social Economic Factors on Landscape Pattern Indices—The Case of the Yellow River Basin in Henan Province

**Suming Ren [1], Heng Zhao [1,2], Honglu Zhang [1], Fuqiang Wang [1,2,3,* and Huan Yang [1,2]**

[1] School of Water Conservancy, North China University of Water Resources and Electric Power, Zhengzhou 450046, China; rsmwsbn@126.com (S.R.); zhaoheng@ncwu.edu.cn (H.Z.); z13303243955@163.com (H.Z.); yanghuan@ncwu.edu.cn (H.Y.)
[2] Henan Provincial Key Laboratory of Water Resources Conservation and Intensive Utilization in the Yellow River Basin, Zhengzhou 450046, China
[3] Henan Provincial Key Laboratory of Hydrosphere and Watershed Water Security, Zhengzhou 450046, China
* Correspondence: wangfuqiang@ncwu.edu.cn

**Abstract:** The Yellow River Basin holds significance as a vital ecological shield and economic hub within China. Adapting land utilization practices and optimizing landscape patterns are of paramount significance in preserving the ecological equilibrium of the Yellow River Basin while fostering high-quality economic development. In this study, we selected the Yellow River Basin in Henan Province as our research area. We use a land use transition matrix and FRAGSTATS 4.2 software to analyze changes in land use and landscape patterns within the study area from 1990 to 2020. Furthermore, Geographical Detector is employed to explore the impact of different natural and social economic factors that have influenced the progress of the landscape surface pattern in the study area. Finally, to identify the zonal aggregation effects of primary components in connection with landscaping feature indices at the city dimension, we use bivariate local spatial autocorrelation. The results are as follows: (1) In terms of land use change characteristics, the area of cultivated land, grassland, shrubs, and bare land shows a decreasing tendency, the area of construction land and forest land shows an increasing tendency, and the water area fluctuates and changes. Most of the cultivated land is shifted to construction land, followed by forest land, construction land, and cultivated land mainly transferred from grassland. (2) At the level of type in terms of shifting landscape patterns, cultivated land, forest land, water, and construction land have a more complex landscape shape, reduced fragmentation, and better natural connectivity. At the overall level, the overall landscape pattern indices are relatively stable, with more patch types and a more balanced distribution. (3) The findings regarding influencing factors reveal that the primary industry output value, population, secondary industry output value, and temperature are the principal driving forces behind the progress of the landscape surface pattern. The main drivers have changed over time in different regions. As indicated by the findings from bivariate local spatial autocorrelation analysis, at the city scale, the leading cause of landscape fragmentation in Luoyang is the primary industry output value, while in Xinxiang, landscape fragmentation is primarily driven by the secondary industry output value and temperature. In this study, we introduce the bivariate local spatial autocorrelation method to analyze the clustering effects of key influencing factors and landscape patterns at the city scale. This is crucial for the harmonized growth of land use planning and the urban economy in the Yellow River Basin.

**Keywords:** land use transfer matrix; GeoDetector; landscape pattern change; driving factors; Yellow River Basin in Henan Province

## 1. Introduction

The Yellow River Basin plays a vital role as an essential barrier and a strategic focal point for agricultural advancement within the confines of China. The progression of China's

economy and society and the maintenance of ecological security depend critically on it [1,2]. As a consequence of the swift advancement of society and urbanization, human endeavors have ushered in substantial modifications to the land utilization patterns of the Yellow River Basin [3]. These changes have had a substantial influence on the basin's ecosystem and have caused the deterioration of farmed land, the quick growth of metropolitan areas, and significant shifts in the layout of the landscape [4,5]. Numerous studies have shown that human activities are changing landscape patterns by changing land use types [6–8]. Land use is considered as the basis for studying changes in landscape patterns. The composite effect between lifestyle choices and the environment becomes apparent through transformations in land use categories [9]. Research on land use/cover change has focused on four main areas: land use change processes [10], driving mechanisms [11], impacts on the ecosystem [12], and projections of future land use change [13,14]. A landscape pattern is a spatial system composed of various landscape elements, and it is an important indicator of the balanced relationship between social and economic development and natural ecology [15]. In some ways, the layout of landscapes reflects how rationally land has been used [16]. The temporal dimension of landscape evolution is the change in the nature of the land surface over time, as expressed through the transformation of land types into each other, and the spatial dimension is the detection of spatial structural composition and structural characteristics using landscape pattern indices [17–19]. In terms of research methods, scholars mostly use land use transfer matrix, diffusion equation, and geographic information map methods to analyze the land utilization or coverage change process from the perspective of temporal change [20–22]. The Markov model has a significant advantage in long-term quantitative estimation [23]. The FLUS model can effectively solve the problem of mutual transformation of the layout of the landscape based on a combination of human and natural causes [24]. Therefore, in order to objectively examine the regional and time-dependent trends of land use, Wang et al. simulated dry land usage depending on the LR-CA-Markov coupled model and FLUS model [25]. The support vector machine (SVM) method has also been used to study land utilization or coverage changes by classifying lands in the landscape and analyzing land utilization or coverage changes based on the classification results [26]. On this basis, the CA-ANN technique is used to predict future land application transformation [21].

A landscape pattern index is a quantitative measure of a spatial pattern [23]. Dhanaraj [27] used the index method to explore the evolutionary attributes of urban landscape patterns rooted in the tenets of landscape ecology. Employing the landscape shape index and Shannon diversity index affords a nuanced examination of the multifaceted dimensions of human activities in terms of aggregation, connectivity, diversity, and density [28–30]. In terms of research objects, the evolution of landscape patterns in parks, woodlands, wetlands, alpine plateaus, and cities has been studied. For example, Cheng [31] found that the type of natural landscape surrounding the park promoted the cooling effect of the park. Using a comprehensive set of landscape indicators, Palmero-Iniesta [32] explored the link between forest cover and changes in the spatial structure of European landscapes. Evans [33] correlated landscape pattern indices with anthropogenic factors to analyze wetlands in Alberta. Zhao [34] used remote sensing imagery to discern and categorize land use typologies in Tianjin and proposed a least-cost model to construct an ecological network in Tianjin, grounded upon the computational outcomes of landscape pattern indices.

Driving force analysis can further identify the influences that cause landscape patterns to evolve and help clarify the causes of landscape pattern changes [35]. Conducting a driver analysis on landscape pattern indices frequently serves as a means to scrutinize the landscape aggregation, degree of landscape fragmentation, and landscape diversity. By examining 30 years of data, Ma et al. found that drivers in the Long Lake watershed were positively correlated with LSI, ED, and SHDI and that landscape fragmentation and diversity increased in the watershed [36]. Analyzing the relationship between landscape pattern indices and drivers from the perspective of urbanization, it was found that topography limits the patch size of the built-up area in Chongqing, making its patch shape simpler [37].



By linking landscape patterns to drivers, the role of different types of drivers in influencing landscape aggregation, fragmentation, and diversity can be seen more clearly. And the selection of drivers focuses on quantitative and qualitative analysis of anthropogenic and natural factors [38]. Some scholars used complex networks to analyze the interactions of landscape pattern indices. The relationship between landscape patterns and influencing factors was also examined using geographically weighted regression [39]. The evolution of landscape patterns is an intricate process, shaped by the interplay of natural and social economic factors. Analysis of land use/cover trends in Ethiopia from 1973 to 2019 through satellite imagery and multiple interviews found that analyzing population growth and policy changes can lead to land degradation and have an inverse impact on land change [35]. Other studies assessed the impact of the combination of population, topography, climate, and geographical positioning on the shift of land use types in Rwanda region [38]; urbanization and population growth's effects on land use alterations in Arizona [40]; the impact of economic growth rate, population growth rate, and policies to protect the environment on the land use change in Pakistan [41]; and the impacts of social economic factors on river and lake wetland ecosystems using principal component analysis on wetland areas such as lakes and rivers in Xiong'an New Area wetlands [42]. Research has also shown that the urbanization rate, fixed asset investment, and the degree of pollution have a great impact on land cover change [43]. Previous studies on the factors influencing landscape pattern evolution have shown that the predominant forces driving this evolution are anthropogenic, with a strong emphasis on socio-economic elements [44–46]. The pivotal role of human-induced land use change in shaping landscape heterogeneity cannot be ignored. It significantly impacts ecological processes within the layout of the landscape by reshaping the interaction between human activities and the surrounding ecosystem [47,48]. Bivariate spatial autocorrelation has been used to study the spatial correlation of influencing factors. Through spatial autocorrelation modeling, Huang et al. [49] analyzed the relationship between influencing factors, such as the intensity of human activities in the Three Gorges Reservoir area, and the landscape pattern and found that the distance was the key influencing factor affecting the evolution of the ecosystem. Ding et al. [50] used a hierarchical approach of bivariate spatial autocorrelation to study the spatial autocorrelation and spatial heterogeneity of the Liuxi River, effectively capturing the spatial explanatory power of the drivers. In Henan Province, the topography is complex, stretching from the rugged mountains and hills in the west to the expansive plains in the east. Both natural and social economic factors impact the landscape pattern. Current research on the landscape pattern of the Yellow River Basin focuses on the impact of rapid urbanization on the landscape pattern, while there are fewer studies on the landscape pattern from the perspective of agricultural production.

The Yellow River Basin in Henan Province is located in the central hinterland of China. As human activities intensify, the Yellow River Basin finds itself at a critical juncture, navigating the delicate equilibrium between ecological preservation and the pursuit of rapid economic development. The first decade of the 21st century was the fastest decade of urbanization in Henan Province. The area of agricultural land has been significantly reduced, and the grassland ecosystem has been significantly degraded. The continuous increase in urban construction land area persists [51–53]. Therefore, in order to solve complex land use problems, the impact of urbanization alterations on landscape patterns is analyzed. The core aim of research in Henan Province, renowned for its substantial agricultural sector, revolves around manipulating the intricate dynamics among agricultural production, urbanization, and ecological conservation. Hence, it is imperative to conduct a thorough analysis of the evolving landscape patterns within Henan Province's Yellow River Basin, considering the intricate interplay of natural and social economic factors. This study is centered on the Yellow River Basin in Henan Province, analyzing the dynamic change patterns from 1990 to 2020. And based on the theory of landscape ecology, we delve into the changing landscape patterns within the Yellow River Basin. This study employs Geographical Detector to analyze the impact of both natural and social economic

factors. It uses bivariate local spatial autocorrelation to identify the spatial clustering effects among key drivers and landscape pattern indices at the city scale, accurately grasp the developmental trends in the landscape pattern of the Yellow River Basin, and clarify the intrinsic connection between the natural and social economic factors and the landscape pattern, which is of far-reaching significance to the maintenance of the basin's environment and benign economic and social coordinated development.

## 2. Data and Methodology

### 2.1. Study Area

The Yellow River Basin in Henan Province is situated away in the heart and parts below of the Yellow River (110°21′~116°06′ E, 33°37′~36°05′ N). The area is 36,200 km$^2$, accounting for 21.7% of the total area of the Henan Province, and the main tributary rivers include the Yiluo River and the Qin River (Figure 1).

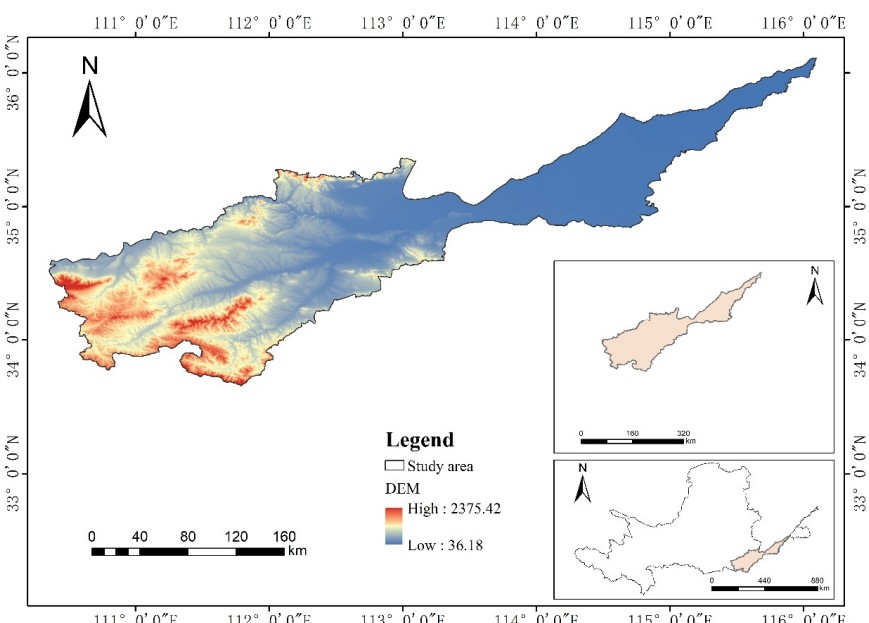

**Figure 1.** Overview of the Yellow River Basin in Henan Province.

### 2.2. Data Sources and Processing

The main data include remote sensing satellite images, DEM data, vector boundaries of the study region, administrative boundary data of Henan Province, and social economic statistics of the study region. The study region's vector boundary and the administrative limits of Henan Province have been sourced from the Data Center for Resource and Environmental Sciences at the Chinese Academy of Sciences (https://www.resdc.cn (accessed on 27 November 2023)). We acquired remote sensing image data from the Geospatial Data Cloud website, managed by the Computer Network Information Center of the esteemed Chinese Academy of Sciences (http://www.gscloud.cn/ (accessed on 27 November 2023)). Social and economic statistics of Henan Province were derived from the Henan Provincial Statistical Yearbook and the Henan Provincial Statistical Bulletin on National Economic and Social Development (https://tjj.henan.gov.cn/ (accessed on 27 November 2023)). In order to make the data sources consistent, the remote sensing images of the study region in 1990, 2000, 2010, and 2020 were used Landsat series satellite data.

ENVI5.3 software was used in this study for pre-processing of remote sensing data, supervised image classification, and the rigorous assessment of the accuracy of classification results. Considering the unique characteristics of the study region and adhering to standardized land use classification criteria, the land use categories were classified into cultivated land, forest land, shrubs, grassland, water, bare land, and construction land (Figure 2).

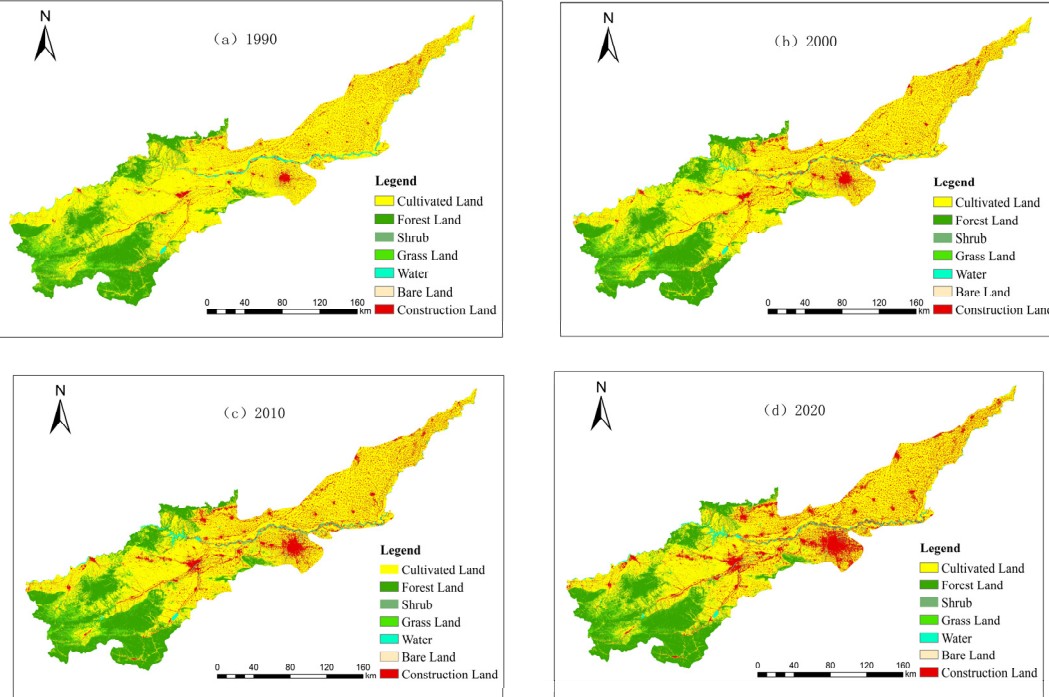

**Figure 2.** Land use distribution in the study region from 1990 to 2020.

*2.3. Study Methods*

2.3.1. Land Use Transfer Matrix

The land use matrix not only reflects the composition of land use in the study area, but also shows the loss and transformation between land use types concretely. To conduct an in-depth investigation into the shifts in land use patterns within the study region, this study utilizes a land use transfer matrix to analyze the dynamics of land use changes in the region [54–56]. Its mathematical expression is as follows:

$$S_{ij} = \begin{bmatrix} S_{11} & S_{12} & \cdots & S_{1n} \\ S_{21} & S_{22} & \cdots & S_{2n} \\ \vdots & \vdots & \vdots & \vdots \\ S_{n1} & S_{n2} & \cdots & S_{nn} \end{bmatrix} \tag{1}$$

In Formula (1), the area of the landscape is included. *n* is the number of land use types. *i* and *j* are the corresponding landscape types before and after land type conversion, respectively.

2.3.2. Landscape Pattern Index

A landscape pattern index is a metric used to quantify landscape characteristics for describing alterations in landscape patterns [57]. Different landscape pattern indices are highly condensed landscape information, and they have some similarities. Therefore, this study combined the actual situation of the study region, following the principle of comprehensively responding to the changes in the landscape pattern while avoiding duplicate selection of landscape indicators. Ten landscape pattern indices were selected from four dimensions: area, shape, dispersion, and diversity (Table 1).

**Table 1.** Describing the landscape pattern index of the Yellow River Basin in Henan Province.

| Index | Expressions | Unit | Applied Scale | Ecological Meaning |
|---|---|---|---|---|
| Percentage of Landscape (PLAND) | $PLAND = \frac{\sum\limits_{j=1}^{n} a_{ij}}{A}(100)$ | % | Patch class | An index that measures the components of the landscape. This value represents the proportion of the patch category area in relation to the entire landscape area. |
| Edge density (ED) | $ED = \frac{E}{A}10^6$ | m/ha | Patch class/landscape | An index that measures the landscape edge parameters. This value represents the edge length between various patch types within a given unit area. |
| largest patch index (LPI) | $LPI = \frac{Max(a_1, \ldots, a_n)}{A_i} \times 100$ | % | Patch class/landscape | This index delineates the attributes of a particular patch type and quantifies its significance within the overall landscape. It is measured as the proportion of the entirety of the region that the largest cluster of that sort covers. |
| number of patches (NP) | $NP = N$ | Pcs | Patch class/landscape | This indicator serves to illustrate landscape heterogeneity. It is determined as the sum of all patches in the overall landscape and the complete number of regions of a certain patch type for the landscape type degree. |
| Landscape Shape Index (LSI) | $LSI = \frac{0.25E}{\sqrt{A}}$ | None | Patch class/landscape | This index indirectly defines the shape attributes of the landscape by measuring the extent to which a patch's shape deviates from that of a circle or a square with the same area. This measurement serves as an indicator of patch irregularity. |
| Aggregation Index (AI) | $AI = \left[\sum\limits_{i=1}^{m}\left(\frac{g_{ii}}{\max g_{ii}}\right)p_i\right] \times 100$ | % | Patch class/landscape | This indicator describes the degree of landscape element aggregation. |
| patch density (PD) | $PD = \frac{N}{A}$ | Pcs/hm$^2$ | Patch class/landscape | This indicator describes landscape heterogeneity, equal to the value of the certain landscape type patch number over the landscape number at the type level, and the value of the overall landscape patch number over the total area at the landscape level. |
| patch cohesion index (COHESION) | $COHESION = \left[1 - \frac{\sum_{i=1}^{m}\sum_{j=1}^{n} P_{ij}}{\sum_{i=1}^{m}\sum_{j=1}^{n} P_{ij}\sqrt{a_{ij}}}\right]\left[1 - \frac{1}{\sqrt{A_i}}\right]^{-1} \times 100$ | % | Patch class/landscape | This indicator describes the overall landscape, reflecting the degree of the spatial layout aggregation. |
| Splitting Index (SPLIT) | $SPILT = \frac{A^2}{\sum\limits_{j=i}^{n} a_{ij}^2}$ | None | Patch class/landscape | This index describes the separateness of landscape patches, equal to the sum of squares of the landscape area divided by the square of all patch areas. |
| Shannon's diversity index (SHDI) | $SHDI = -\sum\limits_{i=1}^{m} p_i(\ln p_i)$ | None | Landscape | A spatial index reflecting changes in landscape abundance and diversity at the landscape level. |

In this comprehensive analysis, we selected a range of landscape indices to evaluate different aspects of the area simultaneously for a degree of specific landscape types and the overall degree. Our chosen metrics included percentage of landscape (PLAND), edge density (ED), largest patch index (LPI), and number of patches (NP) for assessing the overall landscape area. To measure shape regularity and complexity, we utilized the landscape shape index (LSI). Furthermore, we employed the aggregation index (AI) and patch cohesion index (COHESION) to determine the aggregation. To assess fragmentation, we utilized patch density (PD) and the splitting index (SPILT). Lastly, Shannon's diversity index (SHDI) was employed to assess the distribution of landscape diversity. This index, which can characterize landscape spatial variations, was chosen as the primary index for spatial distribution characterization in the examination of landscape heterogeneity in space. LPI was mainly selected to analyze landscape area change, LSI to analyze landscape shape complexity, AI to analyze landscape aggregation, PD to analyze landscape fragmentation, and SHDI to analyze landscape diversity.

### 2.3.3. Geographical Detector

This study investigated the environmental and socio-economic forces that have shaped the landscape pattern evolution in the study region. The social and economic elements included population, primary industry output value, secondary industry output value, tertiary industry output value, agricultural investment, and grain output. The natural factors were precipitation and temperature. GeoDetector was used to analyze the relationship between different drivers and the landscape pattern index. The landscape pattern index was used as the dependent variable, while the independent variables consisted of both natural and socio-economic factors. These variables are instrumental in uncovering the key determinants of landscape pattern evolution. GeoDetector is a tool for identifying spatial dissimilarity and unraveling the factors driving it. The effectiveness of an independent variable in explaining the dependent variable is primarily evaluated by analyzing the similarity in their spatial distribution, and this relationship is quantified through the use of the q-value [17,58]. The formulas are as follows:

$$q = 1 - \frac{1}{N\sigma^2} \sum_{i=1}^{L} N_i \sigma_i^2 S_{ij} \tag{2}$$

$$SSW = \sum_{i=1}^{L} N_i \sigma_i^2, SST = N\sigma^2 \tag{3}$$

where $i = 1 \cdots L$ is the classification of the dependent variable or independent variable; $N$ and $\sigma^2$ are the number of units and variance, respectively; $SSW$ and $SST$ are the sum of variances within the layer and the total variance in the whole area, respectively.

### 2.3.4. Bivariate Local Spatial Autocorrelation

Bivariate local spatial autocorrelation is used to study the spatial clustering effects of major landscape pattern indices and drivers on the city scale [59]. The formula is as follows:

$$I_{ij} = Z_{xi} \sum_{j=1, j \neq i}^{n} \left( w_{ij} \times Z_{yj} \right) \tag{4}$$

where $I_{ij}$ is the local Moran's I; $x$ and $y$ are two variables of region $i$ and region $j$, respectively; $Z_x$ and $Z_y$ are the standardized z-scores for variables $x$ and $y$, respectively. $w$ is the weight matrix. The first-order queen neighbor matrix was used in this study.

## 3. Results

### 3.1. Land Use Change Characteristics

Using the spatial analysis function of ArcGIS 10.2 software, we conducted calculations for the area and land use transition matrix spanning the years 1990 to 2020 within the study

region. The graphical representation of these results can be seen in Figures 3 and 4. From Figure 3, the results show that there are complex and diverse conversion relationships among the land use types. When assessing the conversion of different land use types, it becomes manifest that cultivated land undergoes significant transformations, primarily shifting towards construction land and forest land, making up 62.72% and 23.34% of the overall converted area, respectively. Notably, construction land predominantly originates from previously cultivated land, covering an extensive area of 2298.75 km$^2$, which accounts for a significant 94.29% of the total converted area. Forest land transformations primarily stem from cultivated land, with additional conversions occurring from grassland, constituting 53.27% and 33.62% of the total converted area, respectively. Overall, cultivated land makes up the majority, after which is wooded land, construction land, and grassland, while water, bare land, and shrubs have the smallest share. Between 1990 and 2020, there is a remarkable reduction in cultivated land and grassland, resulting in a respective decrease of 10.56% and 47.78% in their area shares. We noted substantial expansions in the areas of construction land and forest land, with respective area shares of 11.35% and 3.73%. In contrast, the decrease in cultivated land area, while evident, is not notably significant in the overall trend. The area of forest land, construction land, and water area exhibited a consistent upward trend, with construction land showing a more substantial increase compared to forest land and water area. Grassland and shrubs showed an overall decreasing trend, with a more pronounced decrease in grassland. The smallest fraction of the land was made up of bare terrain, which had an overall upward tendency.

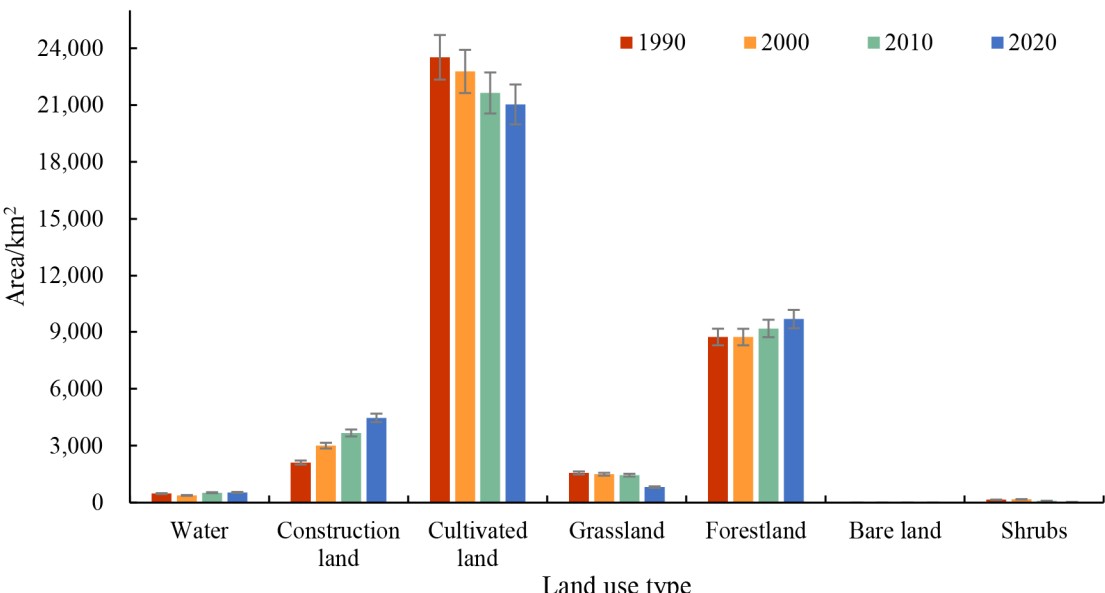

**Figure 3.** Land use area map of the study region.

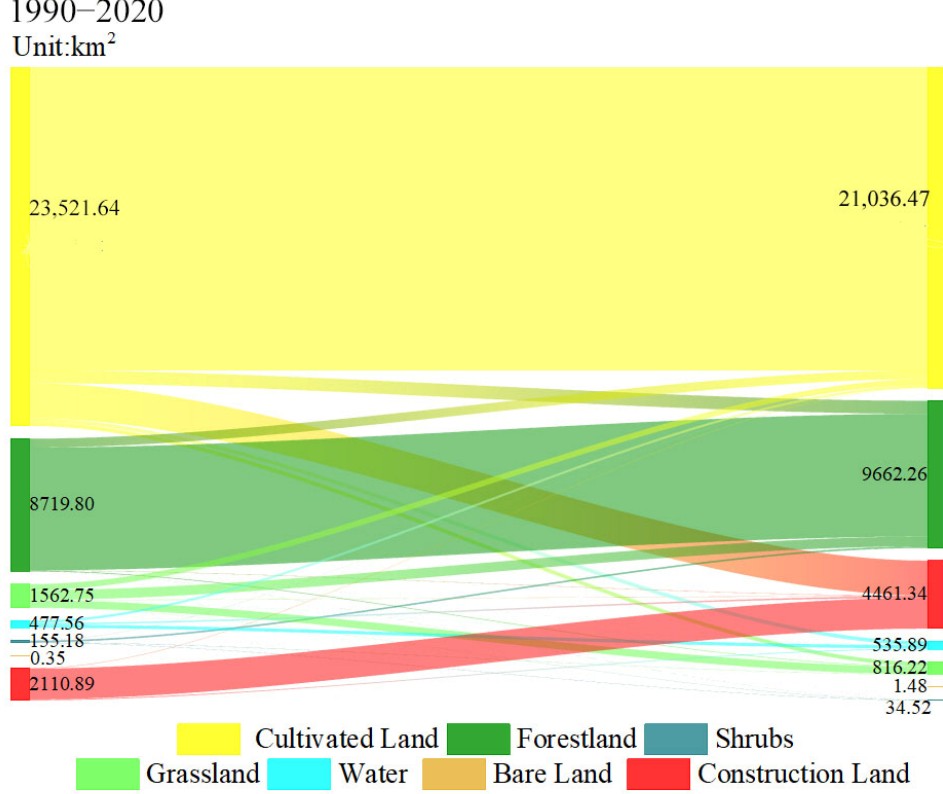

**Figure 4.** Sankey chart of land use type area transfer.

*3.2. Landscape Pattern Evolution Characteristics*

3.2.1. Analysis of Landscape Type Levels

The landscape pattern index is shown in Figure 5. In terms of landscape area shape, it becomes apparent that cultivated land surpasses all other land types with the highest PLAND, underscoring its dominant presence within the landscape. The LPI of cultivated land is the largest among all land use types and shows a decreasing trend. The LPI of grassland and shrubs does not change significantly, and the LPI of watershed shows an up-and-down trend. The ED of construction land is on the rise, whereas the ED of grassland, forest land, and shrubs is declining steadily, and the ED of watershed and cultivated land shows an up-and-down trend. The LSI of each land use type does not differ significantly over the 30-year period, indicating a regular development of the shape of each landscape type. In terms of aggregation and dispersion, the COHESION values of cultivated land, forest land, water, and construction land are all higher than 93, reflecting that the aggregation of these land types is high and the natural connectivity is good. The highest AI is for cultivated land, and the lowest is for shrubs, reflecting that the landscape dispersion is the least for cultivated land and greatest for shrubs in the study region within the last 30 years. The PD values of cultivated land, forest land, shrubs, and construction land show a decreasing trend, and fragmentation is reduced.

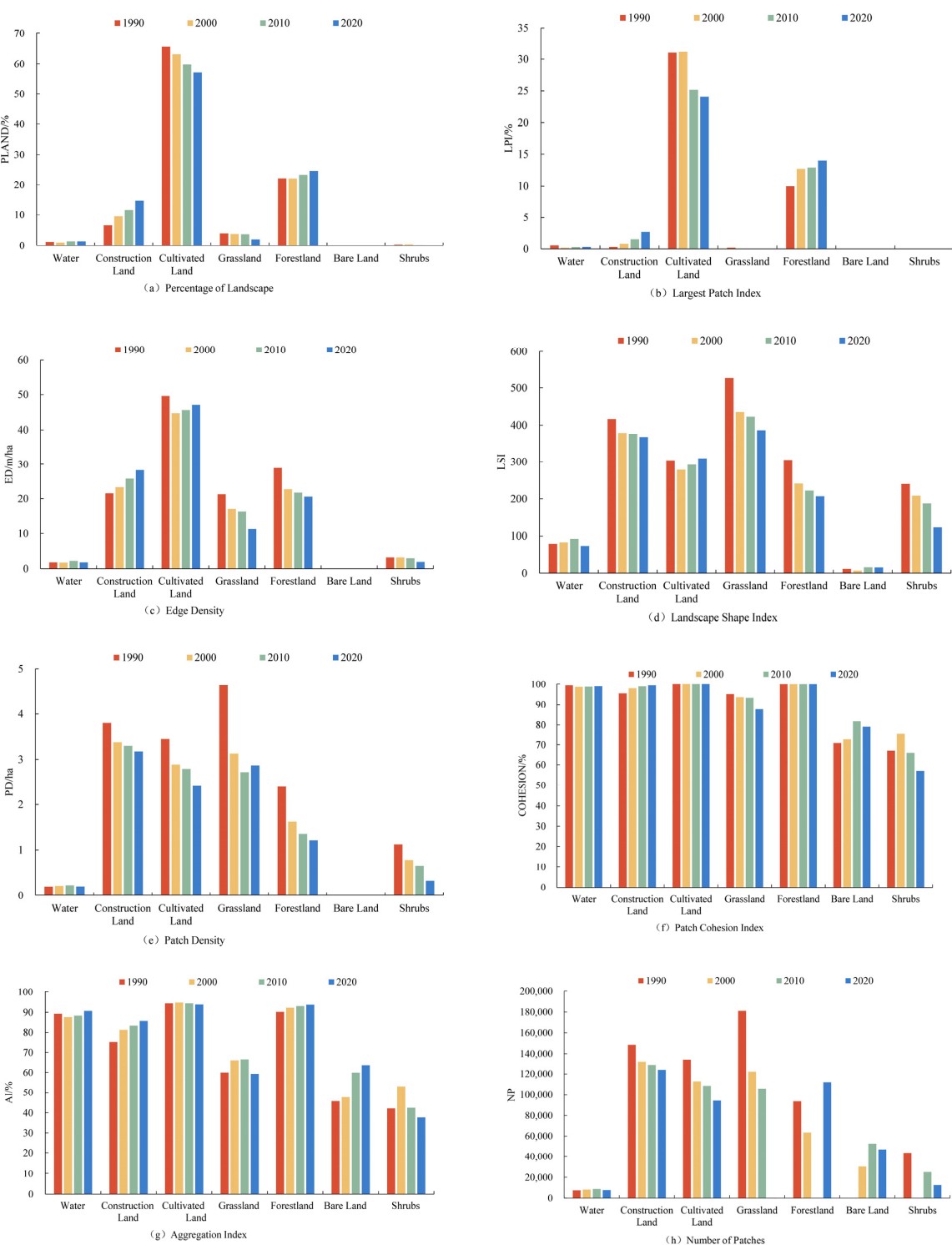

**Figure 5.** Landscape type level index of the study region from 1990 to 2020.

### 3.2.2. Analysis of the Overall Level of the Landscape

The results of the landscape pattern indices are shown in Figure 6, in terms of landscape area shape, the LSI fluctuates and changes from 1990 to 2020, with a slight increase from 2000 to 2010, showing that the landscape area shape of the Yellow River basin change toward irregularity and complexity from 2000 to 2010. Thereafter, there is still a regular trend. The ED is decreasing, indicating that the length of patch edges among different landscape types is gradually decreasing, and the landscape area shape has a trend of

regularization. No significant change in LPI shows that the landscape is more stable internally, and human factors have less interference with the landscape pattern. As can be seen from Table 2, in terms of aggregation and dispersion, the AI values show fluctuating changes, COHESION is around 99, a growing trend is evident in SPLIT, and an overall decrease in PD is reflected by the heterogeneity of the landscape, showing that the landscape integrity has increased, the patch area has decreased, and the fragmentation has weakened throughout the study period. In terms of diversity, the SHDI shows an initial increase followed by a decrease, indicating that the patch types in the study region increase, albeit with an uneven distribution. Nonetheless, the overall balance remains higher than the initial levels observed at the outset of the study.

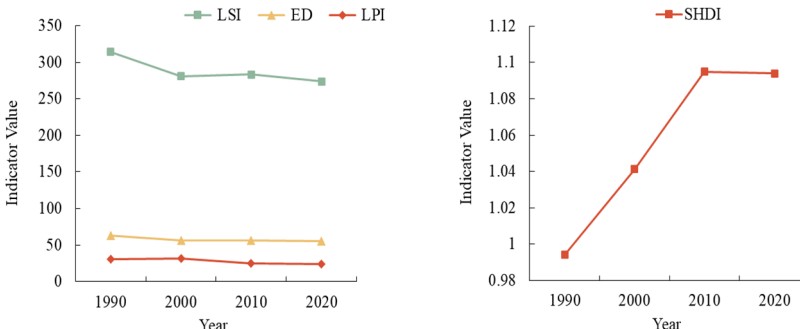

**Figure 6.** Landscape overall level index in the study region.

**Table 2.** Characteristics of horizontal dispersion index in the study region.

| Year | PD | AI | COHESION | SPLIT |
|------|-------|-------|----------|-------|
| 1990 | 15.59 | 90.53 | 99.92 | 5.24 |
| 2000 | 12.00 | 91.54 | 99.92 | 5.45 |
| 2010 | 10.99 | 91.47 | 99.91 | 7.06 |
| 2020 | 10.18 | 91.75 | 99.91 | 7.68 |

3.2.3. Analysis of the Spatial Distribution Characteristics of the Landscape

The results of the spatial distribution of the landscape indices are shown in Figures 7–11. In terms of landscape area shape, LPI values exhibit a west-to-east gradient, with lower values found in specific regions of Lushi County and Lingbao City; the LSI values are high in the southwest part of the basin; and the patch complexity is relatively low in the east compared to the southwest. Cultivated and construction land together form the prevailing landscape in this region. In terms of aggregation and dispersion, the high values of AI are located in the central and eastern areas, where cultivated land is the dominant landscape, whereas lower AI values can be observed in the southwestern part of the region. The AI in the western part is on the rise in 2020, and the level of aggregation in the basin landscape is increasing. The PD shows clear western highs and eastern lows, with greater changes in the regions of the middle and west during the study period, while the eastern region is in a weakened state of fragmentation due to the high protection of the landscape patterns which are mainly cultivated land and construction land. In terms of diversity, the distribution of SHDI maxima is mainly in hilly areas, and the minima are distributed in the eastern region. Over the course of the study, the range of landscape diversity has further expanded and shows the characteristics of banded connectivity.

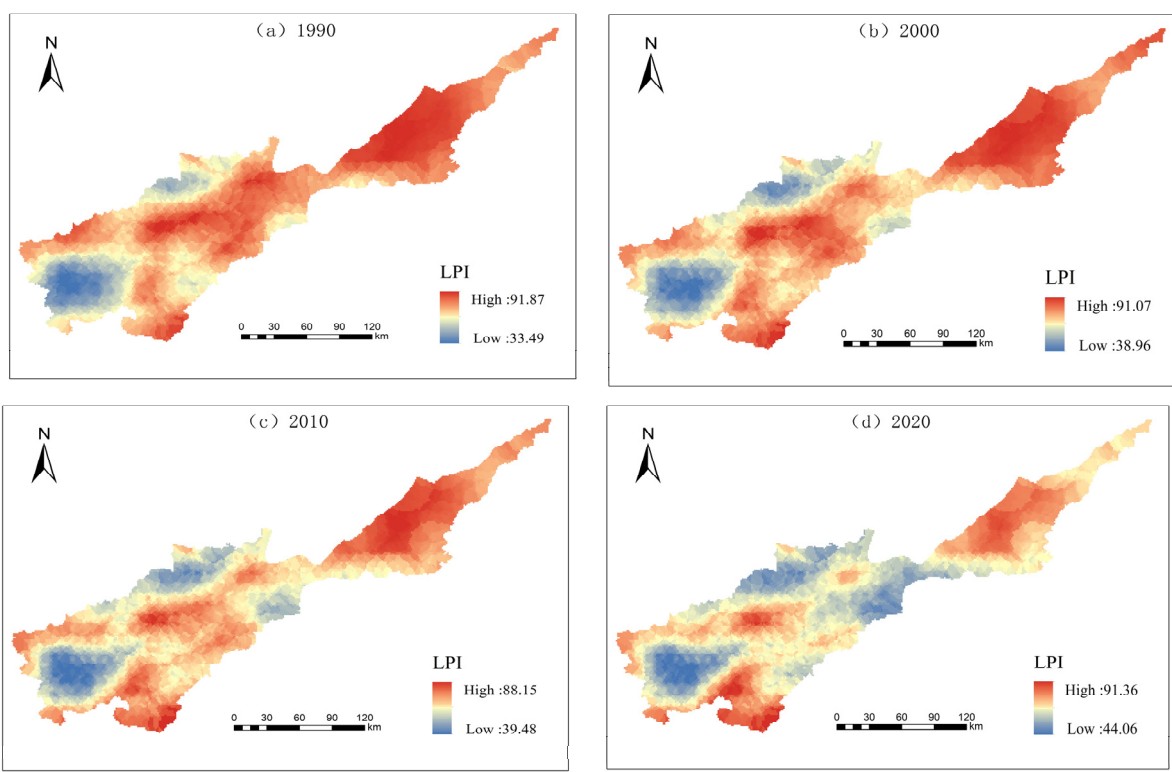

**Figure 7.** Spatial distribution of patch index of maximum landscape type.

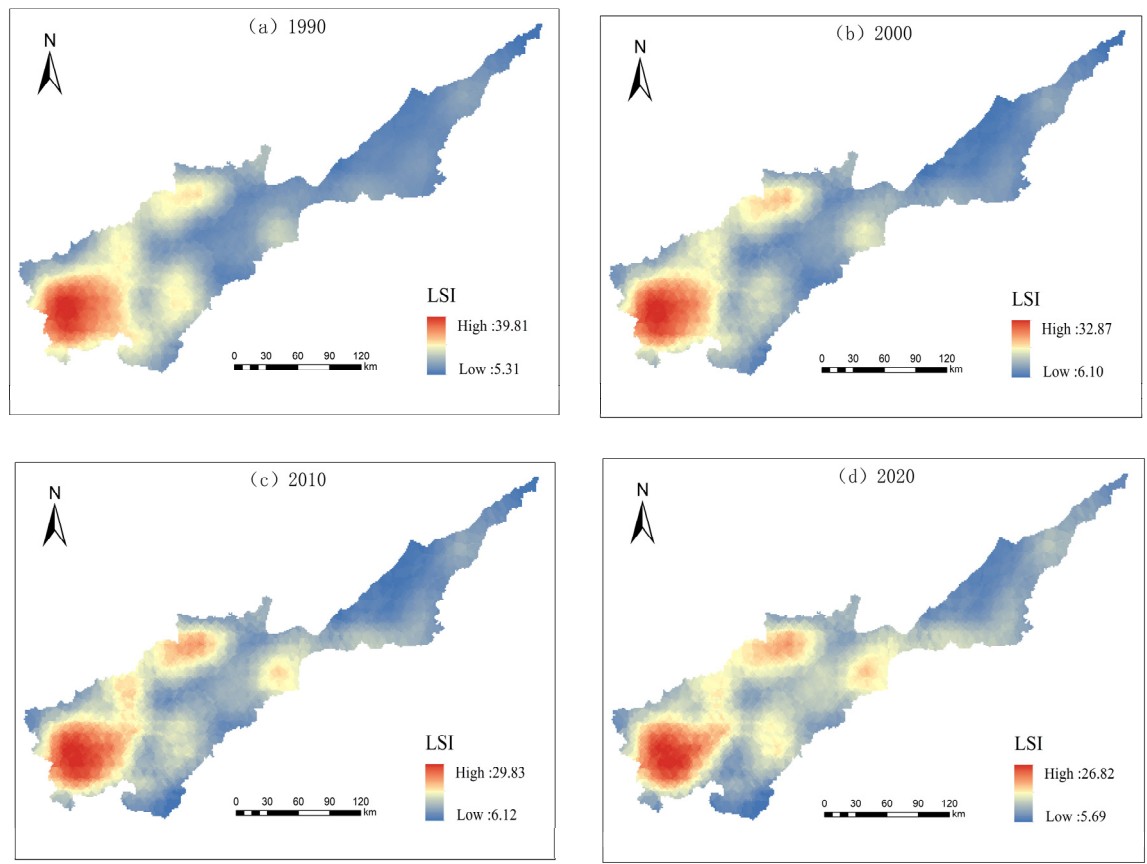

**Figure 8.** Spatial distribution of landscape shape index.

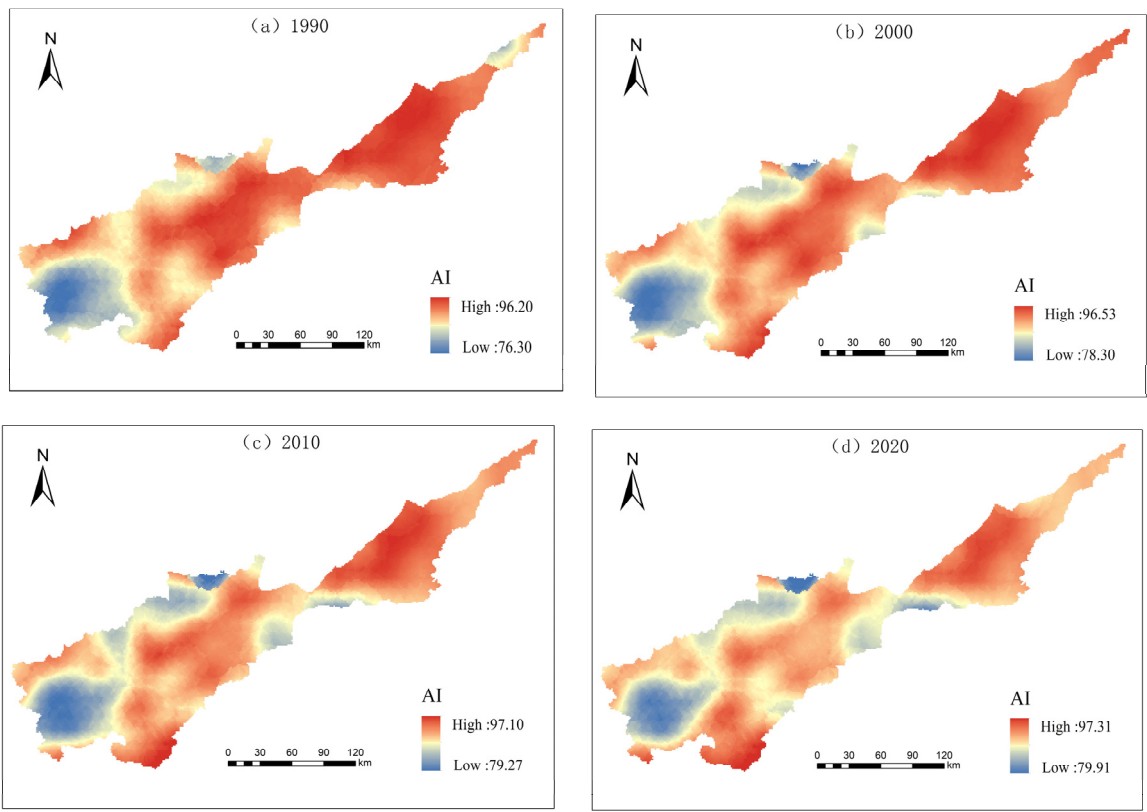

**Figure 9.** Spatial distribution of landscape aggregation index.

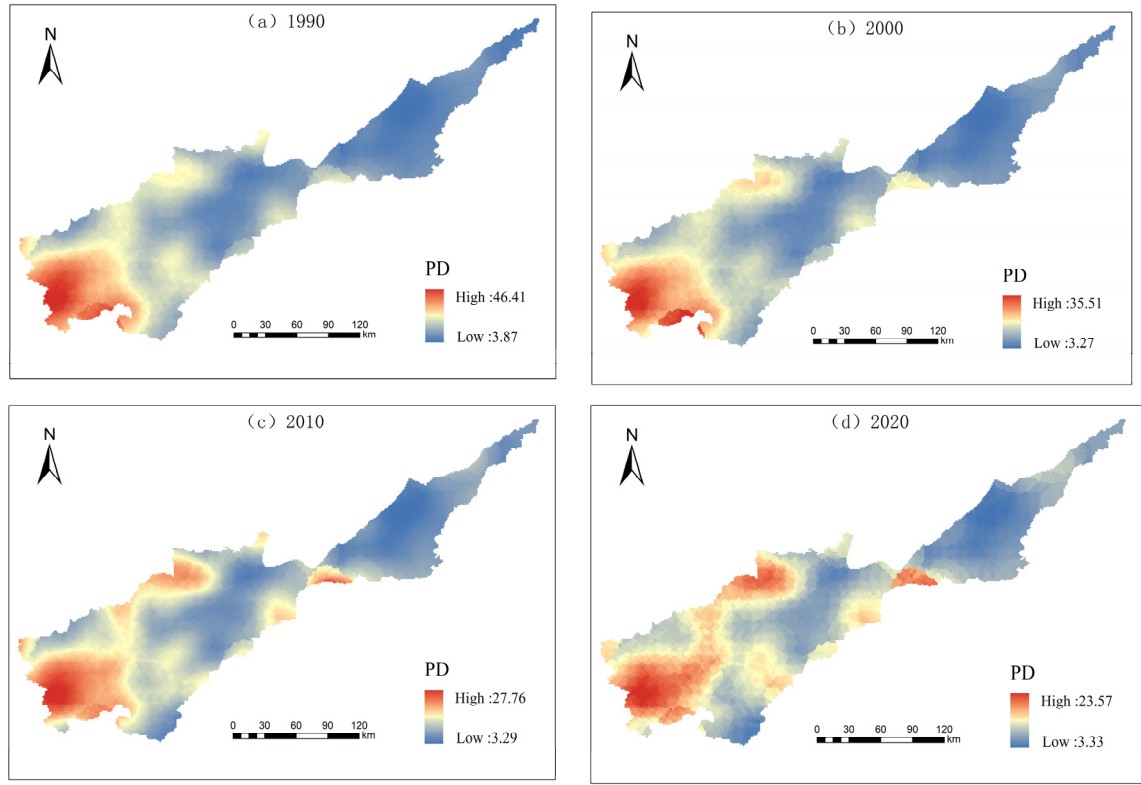

**Figure 10.** Spatial distribution of patch density.

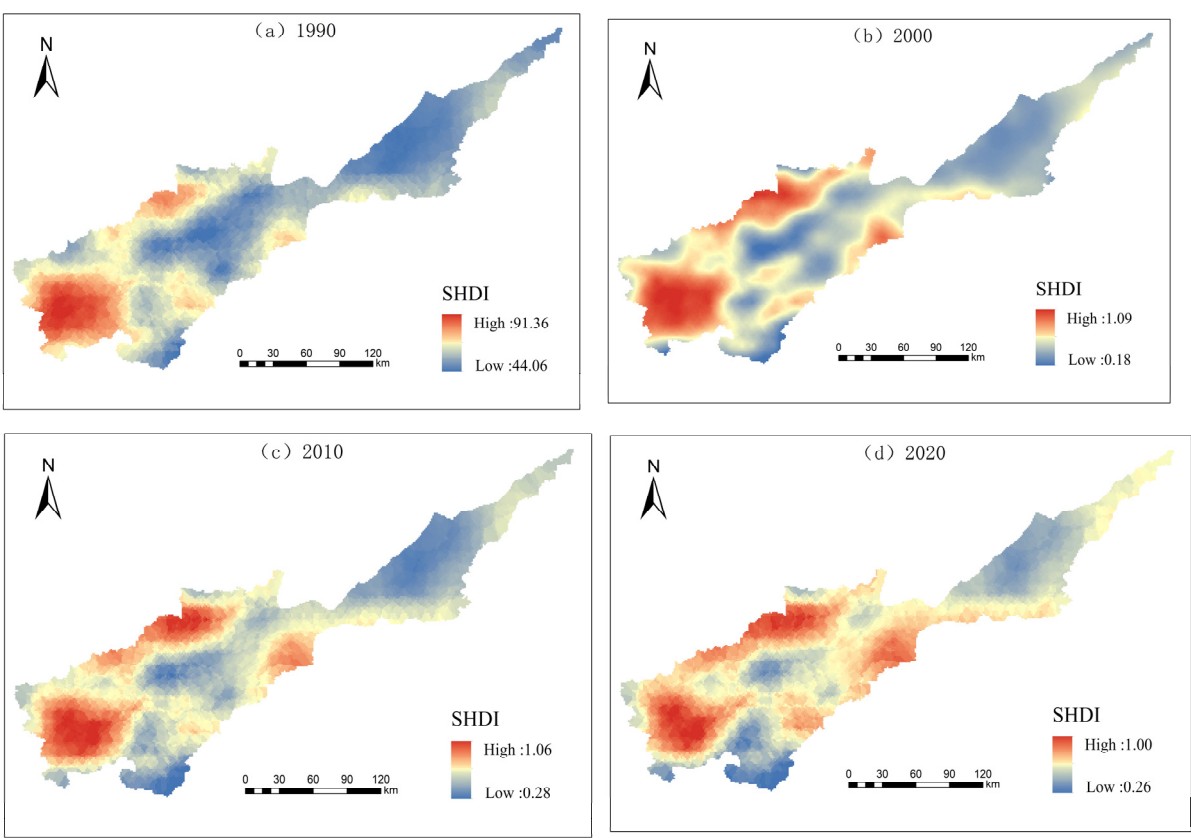

**Figure 11.** Spatial distribution of landscape diversity index.

### 3.3. Driving Factor Analysis

The GeoDetector results showing the influence of different drivers on the landscape pattern index from 1990 to 2020 are shown in Figure 12. Considering different years, the influence of the secondary industry output value on the spatial distribution of landscape pattern indices such as AI, LPI, LSI, PD, and SHDI in 1990 is the largest among the influencing factors, with a mean q-value of 0.53, and the factor with the smallest influence is precipitation, with a mean q-value of 0.27. Temperature has the greatest explanatory power for the geographical distribution of each index in 2010, with a mean q-value of 0.52, and the least influential factor is agricultural investment, with a mean q-value of 0.34. Agricultural investment has the greatest influence on the spatial distribution of the landscape pattern index in 2000 and 2020 among the influencing factors, with a mean q-value of 0.54 and 0.45, respectively. The next largest influencing factors are population and primary industry output value, while precipitation has the lowest explanation capacity with mean q-values of 0.35 and 0.29, respectively. In addition, a comparison of the q-values over four years shows that different influencing factors undergo unique changes in different years. In 2000, the q-values of all the influences increase, and the explanatory power increases. In 2010, the q-values of the secondary industry output value continue to increase, and the q-values of all other factors decrease. By 2020, the q-values of the factors continue to decrease, except for the increase in the q-value of agricultural investment. The results of the mean q-value of the detection results of the index of the study region from 1990 to 2020 are shown in Figure 13. The hierarchy of the explanatory power of the impact factors is determined by the mean value of q. The primary industry output value > population > secondary industry output value > temperature > grain output > agricultural investment > tertiary industry output value> precipitation. From the mean value, the primary industry output value, population, and secondary industry output value among social economic factors have a greater influence on the landscape pattern index. And the temperature among natural

factors has a greater influence on the landscape pattern index. See the Supplementary Materials in the annex for a chart categorizing the drivers.

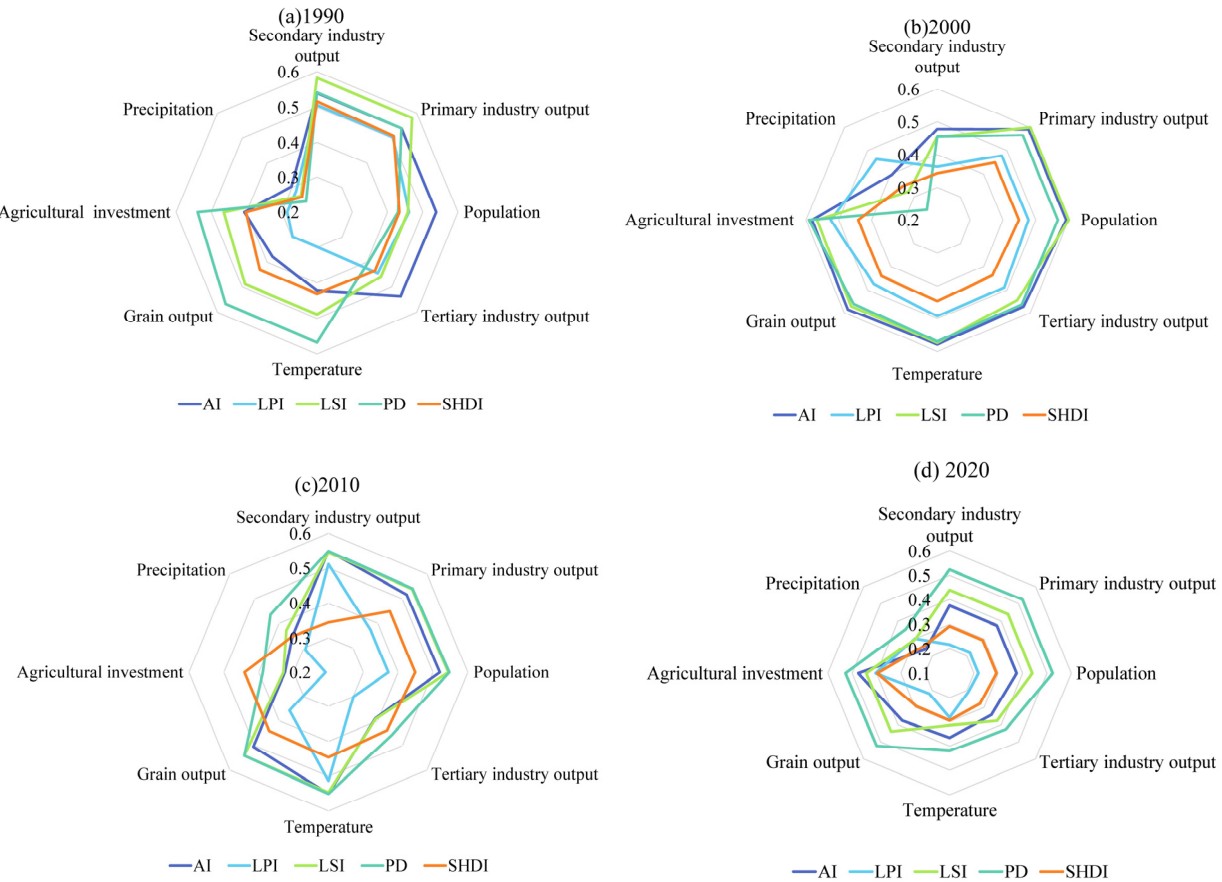

**Figure 12.** q-values of different drivers for different landscape pattern indices.

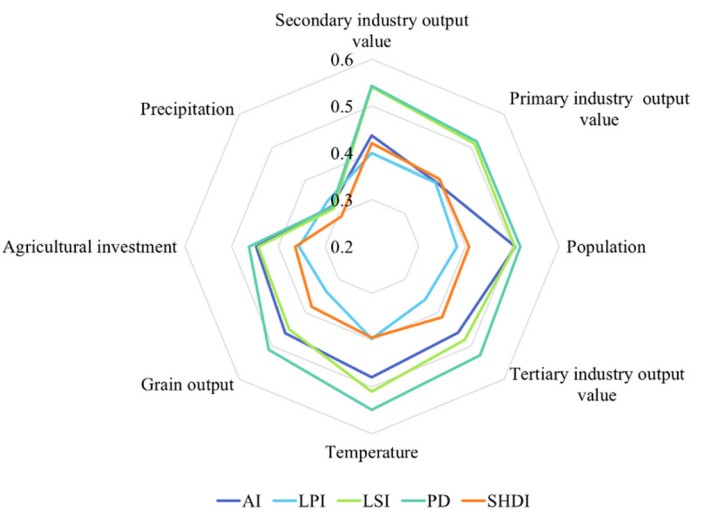

**Figure 13.** Mean values of q-values for the effect of drivers on landscape pattern indices.

Figures 14–17 display the outcomes of the bivariate local spatial autocorrelation analysis. As can be seen from the figure, the spatial correlation between the main drivers and the index is significant in three regions, Luoyang, Puyang, and Xinxiang, and not significant in other regions. This may be due to the fact that the spatial clustering of the drivers and the index is not as pronounced in the other localities compared to Luoyang, Puyang, and

Xinxiang and that the choice of drivers focuses on agricultural development and may be somewhat subjective. So, the lack of significance in the other regions will not affect the correlation between the selected factors and the index. In 1990, the primary industry output value is positively associated with factors like fragmentation, shape complexity, and diversity in Puyang and Luoyang and negatively correlated with the landscape fragmentation in Xinxiang. The connection between population and landscape pattern indices is consistently uniform across all three regions—Luoyang, Puyang, and Xinxiang—as population exhibits a positive correlation with factors such as fragmentation, shape complexity, and diversity. The secondary industry output value is positively associated with landscape fragmentation, shape complexity, and diversity in Xinxiang and negatively connected with landscape fragmentation in Luoyang and Puyang. In the region of Luoyang, we observe an inverse relationship between temperature and landscape fragmentation, shape complexity, and diversity; Xinxiang and Puyang are the opposite.

In 2000, the relationship between the primary industry output value and the landscape pattern indices is the same in Luoyang, Puyang, and Xinxiang; the primary industry output value has a direct relationship with the area and aggregation and an inverse association with fragmentation, shape complexity, and diversity. Population is inversely associated with fragmentation, shape complexity, and diversity in Xinxiang and directly related with fragmentation, shape complexity, and diversity in Luoyang and Puyang. The relationship between temperature, secondary industry output value, and landscape pattern indices is the same as that of the population factor, which might be related to the city's swift progress and growth.

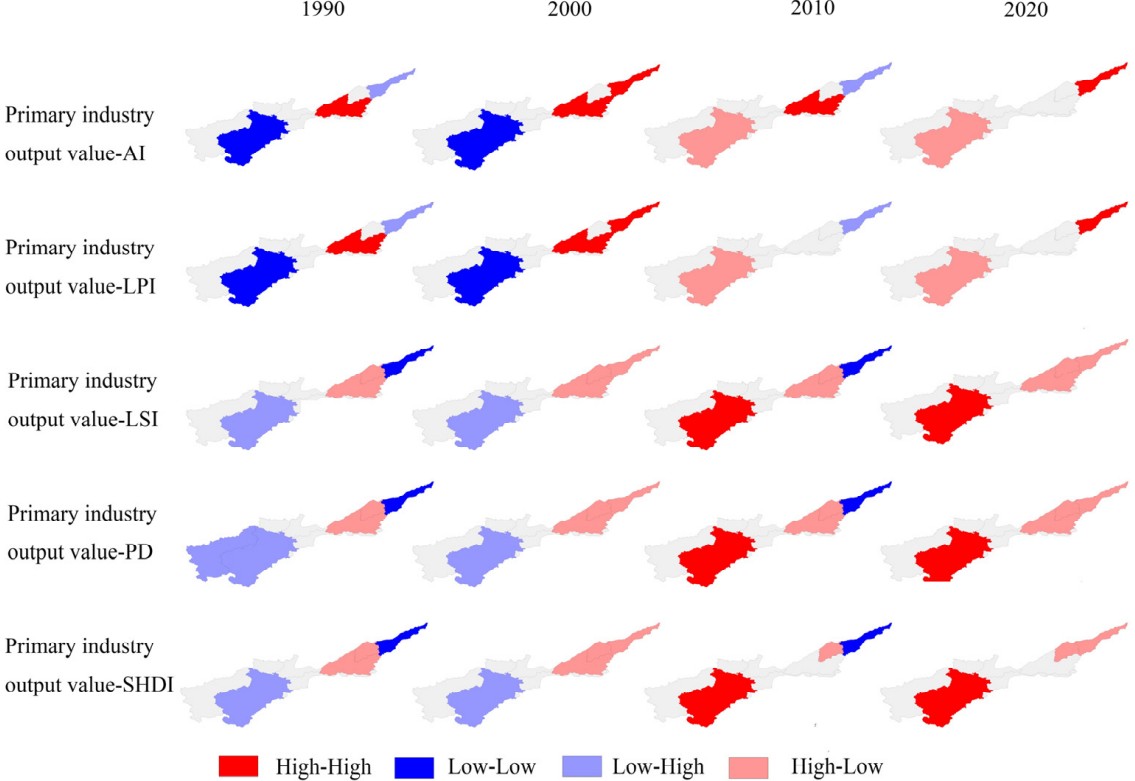

**Figure 14.** Bivariate LISA cluster maps between the primary industry output value and landscape pattern indices.

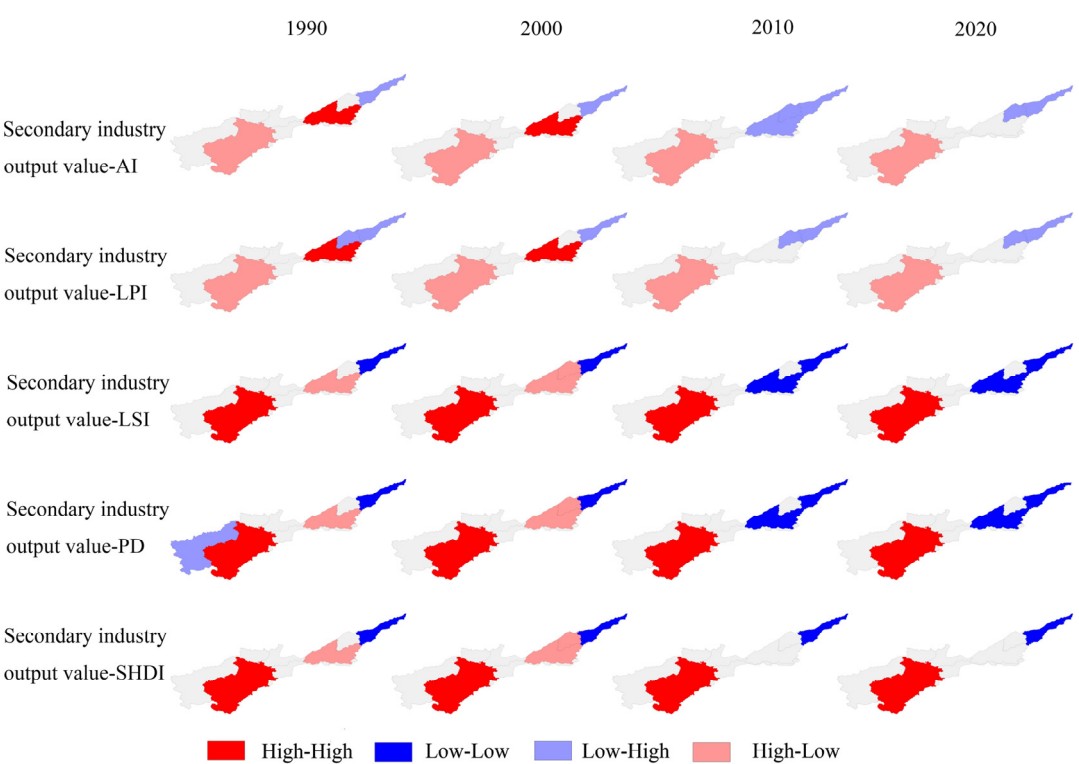

**Figure 15.** Bivariate LISA cluster maps between the population and landscape pattern indices.

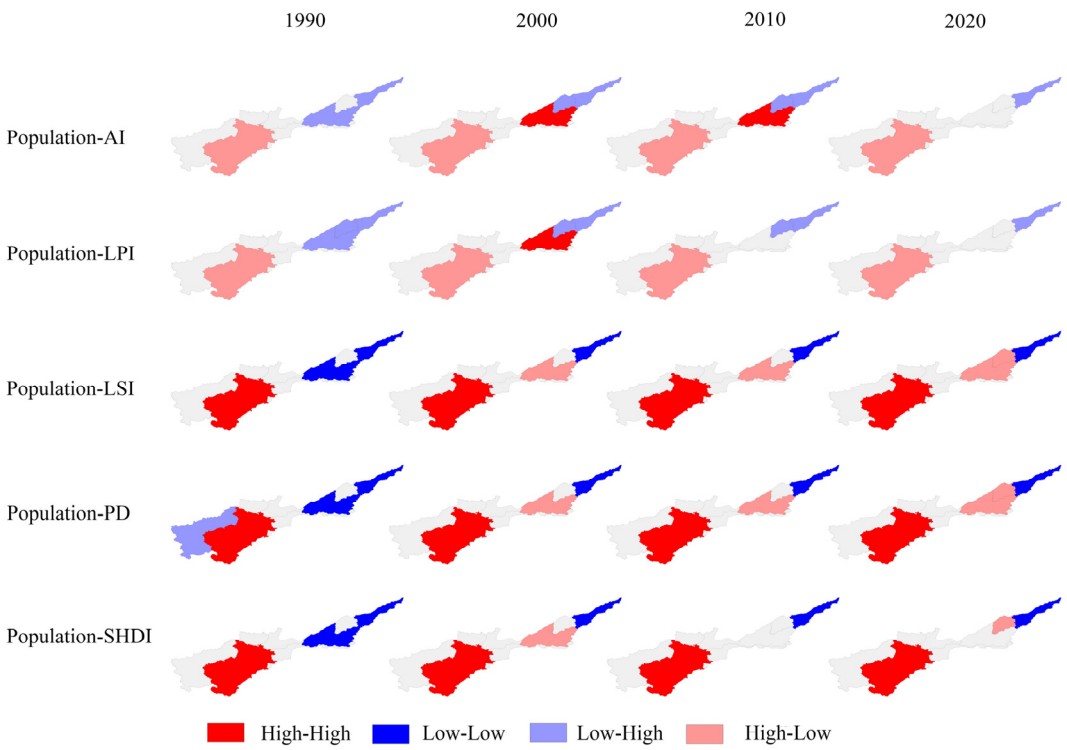

**Figure 16.** Bivariate LISA cluster maps between the secondary industry output value and landscape pattern indices.

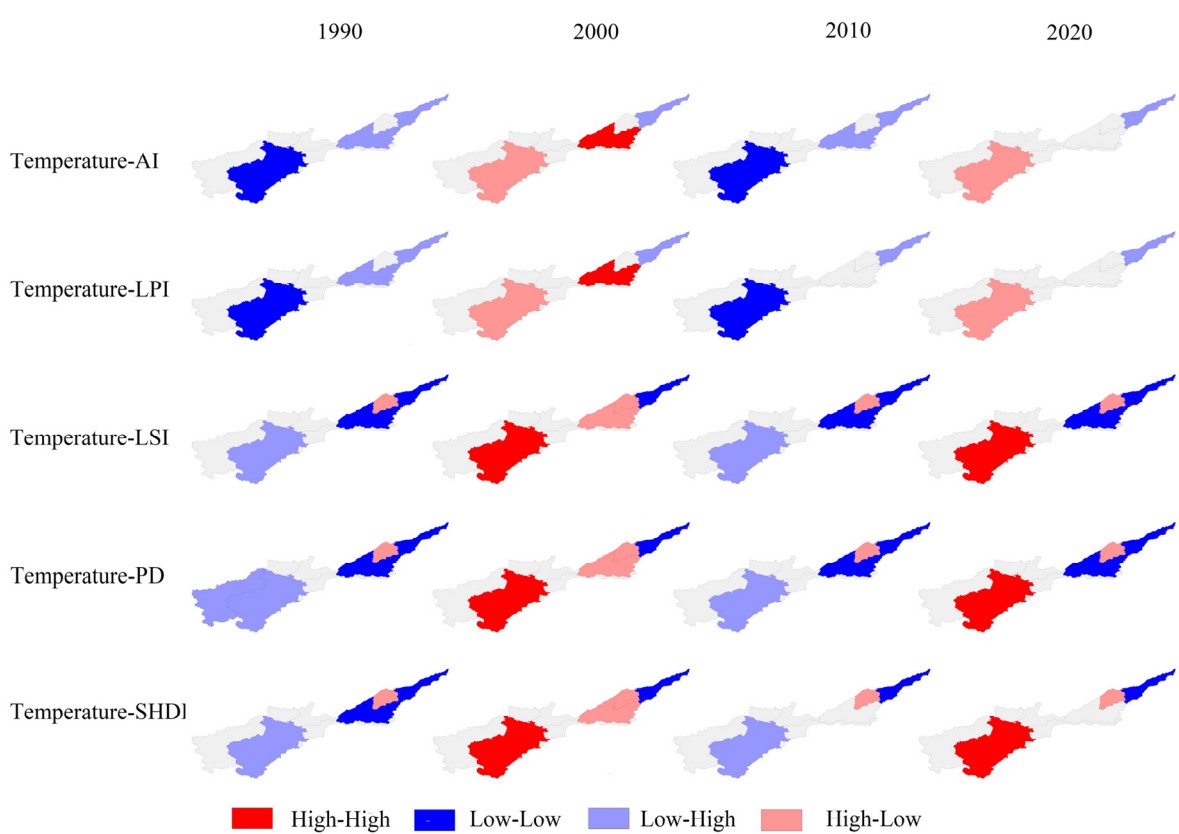

**Figure 17.** Bivariate LISA cluster maps between the temperature and landscape pattern indices.

In 2010, the primary industry output value and population show the same influence on the landscape pattern indices. The primary industry output value and population have an inverse association with the fragmentation in Xinxiang and a direct relationship with the landscape fragmentation and diversity in Luoyang and Puyang. The relationship between the secondary industry output value and the landscape pattern indices is the same in Luoyang, Puyang, and Xinxiang, and the secondary industry output value has an inverse association with the landscape area and aggregation and a direct relationship with the fragmentation, shape complexity, and diversity. Temperature has an inverse association with these characteristics in Luoyang and a direct relationship with these characteristics in Puyang and Xinxiang. In particular, the correlations of the primary industry output value, population, secondary industry output value, and temperature with landscape pattern indices LPI and SHDI in Xinxiang became insignificant.

In 2020, an opposing correlation was observed between the primary industry output value and the landscape pattern indices in both Luoyang and Puyang. The primary industry output value has a direct relationship with fragmentation, shape complexity, and diversity in Luoyang. Conversely, in Puyang, there exists an inverse association between the primary industry output value and these landscape characteristics. The correlations of population, secondary industry output value, and temperature with landscape pattern indices are consistent in Luoyang and Puyang. Population, secondary industry output value, and temperature have an inverse association with landscape area and aggregation and a direct relationship with landscape fragmentation, shape complexity, and diversity. In particular, the correlations of primary industry output value, population, secondary industry output value, and temperature with landscape pattern indices AI, LPI, and SHDI in Xinxiang became insignificant. Combined with the results of 2010, the results show that the primary industry output value and population in Xinxiang have an inverse association with the landscape shape complexity and fragmentation, while the secondary industry output value and temperature have a direct relationship with these landscape characteristics.

In summary, the main drivers are varied across regions over time. The relationship between the primary industry output value in Luoyang and the landscape fragmentation exhibited a positive correlation in 1990 and 2020. The population, secondary industry output value, and temperature in Puyang are negatively correlated with landscape area and aggregation and positively correlated with landscape fragmentation and diversity, showing that the population, secondary industry output value, and temperature play pivotal roles as the key driving factors behind landscape fragmentation in Puyang. The spatial correlations of the primary industry output value, population, secondary industry output value, and temperature with landscape pattern indices AI, LPI, and SHDI in Xinxiang became insignificant in 2020 with the change in time. In 2020, the primary industry output value and population of Xinxiang are negatively correlated with LSI and PD, while the secondary industry output value and temperature are positively correlated with LSI and PD, indicating that the temperature and secondary industry output value have steadily replaced the primary industry output value as the principal driving elements of the landscape pattern indices in Xinxiang.

## 4. Discussion

### 4.1. Spatial and Temporal Evolution Analysis

Over the span of three decades, from 1990 to 2020, the significant changes in land use were chiefly associated with the shift of cultivated land, forest land, and construction land. These transformations are largely a consequence of urban expansion and the irreversible shift of cultivated land into construction areas, leading to a considerable reduction in cultivated land area and presenting formidable challenges for reversion in the study region. This further suggests that the swift surge in construction land caused by expanding cities comes at the detriment of other essential land use categories. This corroborates the observed pattern of land use area changes in accordance with findings from previous research [54]. In 2000, the area of forested land witnessed a decline compared to 1990, primarily driven by deforestation and land reclamation to satisfy the food demand resulting from population growth, further intensifying the development and depletion of forested land. The area of forested land in 2020 has increased compared to 2010, mainly because Henan Province has been carrying out forested land preservation and utilization planning since 2012 and has persistently performed reforestation and aerial-seeding afforestation [55]. From 1990 to 2020, the LPI of cultivated land shows a decreasing trend, and the LPI of constructed land shows a growing tendency, while the PLAND of constructed land shows a continuous increasing trend. To a certain extent, this signifies a downward trend in the dominance of cultivated land within the study area's landscape, while highlighting the growing prominence of construction land. Moreover, comparable results have been documented in other literature, underscoring the ongoing swift progress and growth of cities in the Basin. The dominant pattern in this region is the conversion of diverse land use types into urbanized areas [56].

Spatially, the eastern part of the study region has a single landscape type and a low degree of fragmentation, mainly because cultivated land and construction land are the dominant landscapes in the eastern part of the region. Thus, the eastern part of the region has lower patch complexity and fewer patch types, and the reduction in patches indicates a weakening of the degree of fragmentation. This result is supported by existing research [14]. The central and western regions have a variety of landscape types with increased fragmentation due to urban expansion. This is because the central region is a plain area with flat terrain, and the availability of transportation facilities to some extent leads to landscape fragmentation. It has been shown that the landscape on both sides of highways and railroads has a higher degree of fragmentation [60].

### 4.2. Analysis of Influencing Factors

Based on the results of GeoDetector's influence factor identification, the alteration in the landscape pattern indices is primarily driven by factors such as the output value

of the primary industry, population dynamics, the secondary industry output value, and temperature variations. Related studies have also concluded that alterations in landscape pattern indices are influenced by the primary industry output value and population. Ma concluded that land use and ecological changes predominantly result from various socio-economic activities [36]. Changes in major landscape pattern indices are more sensitive to the effects of the primary industry output value and population than to any other factors. This is in harmony with the outcomes of studies examining the factors that impact changes in landscape patterns in different watersheds [19,61].

The study of 2010 finds that temperature has the greatest influence on landscape pattern index changes. And combined with the results of multi-year averages, temperature has a greater influence on the indices, which differs from the findings of other scholars on the drivers of landscape pattern changes [57,62,63]. Liu also noted that lifestyle choices had an enormous effect on how the landscape has evolved [64]. The rapid urbanization that occurred in Henan Province during the first decade of the 21st century can be identified as the cause of this phenomenon. Research on land use types has additionally revealed that urban development is responsible for the conversion of cultivated land, forest land, and grassland into construction land, corroborating previous findings [65]. Imran also indicated that the increase in built-up land has an inverse impact on temperature [66]. The increase in impervious area leads to an increase in surface temperature. This is consistent with Sun's findings on the landscape pattern and urban heat island effect in Chengdu, where an increase in cold landscapes, such as cultivated land, forestland, and grassland, had a mitigating effect on surface temperature [67]. Chen also emphasized that the adverse effects of urbanization on surface temperature changes are more significant in the early stages of urbanization [68]. Past research has convincingly illustrated that urban areas experience higher temperatures compared to their neighboring rural counterparts [69]. Li's research on Zhengzhou unveiled a robust correlation between temperature and the landscape pattern of high-rise buildings [70]. The conclusion that temperature has a strong influence on landscape pattern indices is therefore reliable. This also implies that in the future, determining how to improve the connectivity of cooling landscapes such as grassland and forestland and to decentralize the spatial layout of construction land is a priority for land planning.

Over the entire period, the drivers that have had the most substantial impact on the landscape pattern index varied from year to year. This paper argues that this is mainly a result of policy regulation. Similar studies have emphasized the impact of government policies on landscape pattern changes over time [71]. Long discovered that the improvement in rural-specific standards following the reform and opening up in 1978 prompted rural residents to build new homes on the fringes of cities and villages [72]. The swift advancement of the secondary industry promoted the encroachment of built-up land into cultivated land and grassland. Therefore, the influence of the secondary industry in 1990 was most likely attributed to the implementation of the reform and opening-up policy. In 1997, the government enacted the cultivated land balance policy so that the transformation of cultivated land in one place would be compensated in other places. This is consistent with Li's findings on the relationship between land planning policies and land use changes in Tianjin [73]. Subsequently, Henan Province transitioned into a period of swift urbanization, marking a significant transformation in its developmental landscape. Temperatures in urban areas increased, and most of the green space was transformed into compact construction land [74]. In 2017, the government enacted a policy to guide the transfer of rural land, which led to higher incomes for farmers and a greater connection between farmed areas, and facilitated the development of large-scale agricultural production using large-scale machinery. In summary, although policy factors are difficult to quantify, they affect the evolution of the landscape pattern to varying degrees. Thus, it is imperative to factor in the preservation of cultivated land when shaping relevant policies, with the aim of mitigating the repercussions of construction land expansion on cultivated land fragmentation and ultimately optimizing land use efficiency.

The significantly influenced landscape fragmentation in Luoyang is shown by the results of bivariate spatial autocorrelation analysis to be the output of the primary industry. This may be because in Luoyang, as a significant industrial hub in the heartland, the loss of arable land is the most common consequence of urban expansion [75]. Huang et al. found that cropland dominated the landscape fragmentation in Luoyang City by analyzing the trend of the landscape pattern of cropland in Luoyang City [76]. Luoyang City's policy of storing grain on the land has greatly guaranteed food security and arable land resources and increased the output value of the primary industry [77]. In 2009, the government put forward the action of "preserving the red line of arable land", and an entire area of agricultural land emerged in Luoyang through measures such as returning land to cultivation for infrastructure and leveling the land. However, the natural connectivity of cultivated land has declined, and cultivated land is distributed in patches. Su's study of rapidly developing areas also shows that urbanization with housing expansion and road construction has altered the cultivated landscapes and that cultivated land in the basin is becoming fragmented [78]. The key driver of the landscape pattern indices in Xinxiang shifted from the primary industry output value to the secondary industry output value. This may be because of the flat topography of Xinxiang, which was dominated by agricultural production activities in the early stages of the city's construction, and with the rapid development of urbanization in Henan Province, Xinxiang's economic focus gradually shifted from agriculture to industry. This is consistent with Luo's research on the process of landscape fragmentation, where industrialization led to agricultural production losing its importance to the economy, and the swift expansion of the secondary industry was the catalyst for landscape fragmentation [79]. Fan's research on Fengqiu highlights the intricate connection between the development of industrial agglomeration and the intricate shapes of landscape patches [80].

*4.3. Limitations*

We analyzed the landscape pattern evolution using land use data over three decades, identified the main drivers of landscape pattern evolution in the Yellow River Basin in Henan Province through GeoDetector, and analyzed the effects of the main drivers on the landscape pattern in different cities. This paper launched a study on the drivers of landscape pattern evolution in the Yellow River Basin of Henan Province at the urban scale and obtained some meaningful research conclusions. Landscape pattern change, however, is a multifaceted process influenced by a combination of factors. The choice of drivers in this paper focuses on agricultural development and fails to demonstrate the impact of urbanization development on landscape patterns. To gain a deeper understanding of the intricate changes in landscape pattern indices, future studies should consider selecting more influential factors to characterize the evolution of landscape patterns at different scales. In addition, we used bivariate local spatial autocorrelation, which only considers spatial agglomeration effects at the city scale and does not consider the influence of neighboring areas. We can change the scale type in future studies to obtain more scientifically sound results.

**5. Conclusions**

In this study, we examined the evolution characteristics of landscape patterns in the Yellow River Basin in Henan Province over three decades and revealed the spatial change patterns. On this basis, GeoDetector was used to identify the influence of natural and social economic factors on the changes in landscape pattern indices in the study region, and bivariate local spatial autocorrelation was used to analyze the spatial clustering effects between the main driving factors and the landscape pattern indices. The main conclusions are as follows:

(1) In the study region, the reduction in cultivated land area and the expansion of construction land area are primarily the result of the transformations of cultivated land and forest land. Policies have increased the area of forest land and turned grass-

land into forest land. Grassland has the highest land use conversion rate, and more grassland area is developed into other land types.

(2)  Landscape types in the southwest are characterized by low connectivity and patch fragmentation, and affected by the urban area, the degree of fragmentation has increased. The eastern region has low SHDI, low patch complexity, and weak fragmentation due to the relative prominence of dominant landscapes.

(3)  The study reveals that within the research area, the primary drivers behind alterations in the landscape pattern indices are the output value of the primary industry, population, output value of the secondary industry, and temperature. At the city scale, the primary industry output value in Luoyang became the primary driver of landscape fragmentation, while the primary factor causing fragmentation in Xinxiang City shifted to secondary industry and temperature. In Puyang, the output value of primary industry, population, output value of secondary industry, and temperature are all positively correlated with landscape fragmentation. Alterations in the indices are mainly related to social economic factors, and the influence of policies on alterations in the landscape pattern indices cannot be ignored. This study can provide a solid scientific basis for the Yellow River Basin's integrated growth in industry and safeguarding of the environment, providing a reference for harmonizing the relationship between the economy and the environment in the Yellow River Basin.

**Supplementary Materials:** The following supporting information can be downloaded at: https://www.mdpi.com/article/10.3390/w15234174/s1.

**Author Contributions:** Conceptualization, writing—original draft, S.R.; writing—original draft H.Z. (Zhang Honglu); writing—review and editing, H.Z. (Heng Zhao); funding acquisition, supervision, F.W.; investigation, supervision, H.Y. All authors have read and agreed to the published version of the manuscript.

**Funding:** The project was financially supported by the National Natural Science Foundation of the People's Republic of China (52279014), Henan Province Key Research and Development and Promotion Project (Science and Technology) (232102320257), and Key Research and Development Program of Ningxia Hui Autonomous Region (2021BEG02012).

**Data Availability Statement:** The data presented in this study are available on request from the corresponding author.

**Acknowledgments:** The authors would like to express their sincere gratitude to the anonymous reviewers for their constructive comments and useful suggestions that helped us improve this study.

**Conflicts of Interest:** The authors declare no conflict of interest.

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
