# Peer review of "Influence of Natural and Social Economic Factors on Landscape Pattern Indices—The Case of the Yellow River Basin in Henan Province"

_water, doi:10.3390/w15234174_

Round 1
Reviewer 1 Report
Comments and Suggestions for Authors
1. Though your literature review is logical and clear, I would suggest the authors to add relevant studies on bivariate local spatial autocorrelation in the introductory section.
2. There has been insufficient discussion of the drivers in Luoyang, which as far as I can tell has a strong industrial base and is also rich in agricultural resources. If the authors think it is relevant, they could consider citing other literature in addition to what they are discussing now to prove that the primary industry is a driver of landscape fragmentation in Luoyang City
3. Authors need to emphasize the practical applications and importance of their research. The authors can further emphasize the innovation and uniqueness of the study in the conclusion section to highlight the authors' contribution. This will help the reader better understand why the study is so important to the field.
Comments on the Quality of English Language
I suggest that the authors need to find professionals with an English background to make appropriate revisions to the quality of this article.
Reviewer 2 Report
Comments and Suggestions for Authors
In this manuscript, the authors attempted to characterize land use change tendencies and landscape patterns as affected by natural and social economic factors in the Yellow River Basin in Henan Province based on a dataset from 1990 to 2020. This study provides an interesting perspective to understand relationships between economic-social development and landscape evolution in the Yellow River Basin. The manuscript is generally well organized and clearly stated. The authors are suggested to carefully check the expressions of the manuscript to avoid grammatical errors and spelling mistakes. Besides, native speakers should be invited to polish the language of the manuscript. Minor revision is suggested and some detailed comments are as follows.
i) In Section 3.3, The outcomes of the bivariate local spatial autocorrelation analysis in the driving factor analysis showed that the spatial correlation between the main driver and the index was significant only in Luoyang, Puyang and Xinxiang, but not in other regions. Additional interpretations are needed to understand that this did not affect the correlation between the selected factors and the index.
ii) Significance and potential practical application of the study could be further emphasized, and some suggestions or policy recommendations could be put forward. Directions of future research and possible improvement in methods could also be proposed.
iii) In Line 205, there was no Table 1 when Table 2 was tabulated.
iv) In Lines 433-435, please confirm whether the relationship between the primary industry output value in Luoyang and the landscape fragmentation presented a negative correlation in 1990.
Comments on the Quality of English Language
The authors are suggested to carefully check the expressions of the manuscript to avoid grammatical errors and spelling mistakes. Besides, native speakers should be invited to polish the language of the manuscript.
Reviewer 3 Report
Comments and Suggestions for Authors
This paper, entitled Influence of Natural and Social Economic Factors on Landscape Pattern Indices-The Case of the Yellow River Basin in Henan Provinc, is a scholarly work and can increase knowledge on this domain. The authors provide an interesting and original study, the content is relevant to Water.
I have some general and specific comments:
- Please provide legends for Figure 4 (Sankey chart)
- why focusing from 1990 to 2020? is there any data before 1990?
- What could be the trends of evolution for the future?
- Please provide error bars for Figure 3. What is the accuracy of these data?
- Please provide signification of the legend for Figure12 and Figure 13 (AI, LPI, LSI, PD, SHDI)
As it, this paper is not fully acceptable for publication and requires some additional data and amendments. I recommend the following decision: ACCEPT AFTER MINOR REVISION.
